# Near-infrared spectroscopy of the placenta for monitoring fetal oxygenation during labour

Katja Ražem[1]*, Juš Kocijan[2,3], Matej Podbregar[4,5], Miha Lučovnik[1,5]

1 Division of Obstetrics and Gynecology, Department of Perinatology, UniversityMedical Centre Ljubljana, Ljubljana, Slovenia, 2 Department of Systems and Control, Jožef Štefan Institute, Ljubljana, Slovenia, 3 School of Engineering and Management, University of Nova Gorica, Nova Gorica, Slovenia, 4 Department of Intensive Internal Medicine, General Hospital Celje, Celje, Slovenia, 5 Faculty of Medicine, University of Ljubljana, Ljubljana, Slovenia

* katja.razem@gmail.com

**Data Availability Statement:** All relevant data are within the paper and its Supporting Information files. Actual recordings can be requested from the corresponding author.

## Abstract

Although being the golden standard for intrapartum fetal surveillance, cardiotocography (CTG) has been shown to have poor specificity for detecting fetal acidosis. Non-invasive near-infrared-spectroscopy (NIRS) monitoring of placental oxygenation during labour has not been studied yet. The objective of the study was to determine whether changes in placental NIRS values during labour could identify intrapartum fetal hypoxia and resulting acidosis. We included 43 healthy women in active stage of labour at term. CTG and NIRS parameters in groups with vs. without neonatal umbilical artery pH $\leq$ 7.20 were compared using Mann-Whitney-U. Receiver-operating-characteristics (ROC) curves were used to estimate predictive value of CTG and NIRS parameters for neonatal pH $\leq$ 7.20. A computer-based statistical classification was also performed to further evaluate predictive values of CTG and NIRS for neonatal acidosis. Ten (23%) neonates were born with umbilical artery pH $\leq$ 7.20. Compared to group with pH > 7.20, fetal acidosis was associated with more episodes of placental NIRS deoxygenation (9 (range 2–37) vs. 2 (range 0–65); p<0.001), higher velocity of placental NIRS deoxygenation (2.31 (range 0–22) vs. 1 (range 0–49) %/s; p = 0.03), more decelerations on CTG (25 (range 3–91) vs. 10 (range 10–60); p = 0.02), and more prolonged decelerations on CTG (2 (range 0–4) vs. 1 (range 0–3); p = 0.04). Number of placental deoxygenations had the highest prognostic value for fetal/neonatal acidosis (area under the ROC curve 0.85 (95% confidence interval 0.70–0.99). Computer-based classification also identified number of placental deoxygenations as the most accurate classifier, with 25% false positive and 93% true positive rate in the training dataset, with 100% accuracy when applied to the testing dataset. Placental deoxygenations during labour measured by NIRS are associated with fetal/neonatal acidosis. Predictive value of placental NIRS for neonatal acidosis was superior to that of CTG.

**Funding:** The author JK acknowledges research core funding No. P2-0001, which was financially supported by the Slovenian Research Agency. The Slovenian Research Agency had no role in study design, data collection and analysis, decision to publish, or preparation of the manuscript.

**Competing interests:** The authors have declared that no competing interests exist.

## Introduction

Intrapartum fetal surveillance is performed to prevent fetal/neonatal hypoxia which may lead to childbirth-related neonatal encephalopathy, cerebral palsy and perinatal death. Monitoring fetal heart rate (FHR) represents an important method for fetal surveillance—cardiotocography (CTG) being the golden standard in developed countries. The interpretation of FHR patterns in combination with uterine activity is key for CTG applicability. However, since the introduction of CTG into clinical practice, there has been no decrease in rates of cerebral palsy and perinatal death, while caesarean section and operative vaginal delivery rates have risen significantly [1]. While normal CTG reliably rules out fetal hypoxia with a negative predictive value of around 90%, abnormal CTG tracings only have a positive predictive value of 50–60% for neonatal hypoxia or acidosis [2–4]. Furthermore, large inter- and intra-observer variability in CTG interpretation has been documented even with experienced clinicians [5–7]. Despite these limitations of CTG, there is currently no other more effective, evidence-based adjunctive method for intrapartum fetal surveillance [8].

Near-infrared spectroscopy (NIRS) enables non-invasive, real-time assessment of tissue oxygenation. It was initially promoted as a brain monitor in cardiac surgery and neonatal intensive care, but its use has been extended to various non-cardiac surgeries and critical care settings [9–11]. NIRS has also been studied as a method for assessing fetal cerebral oxygenation during labour. Studies using special probes placed through the dilated cervix on the fetal head during labour showed that NIRS can identify fetuses at risk for intrapartum hypoxic damage [12–15]. This method is, however, relatively invasive and associated with possible risks of probe placement into the uterus and onto the fetus. Additionally, probes used in these studies are not available for everyday clinical practice and are associated with significant costs. Non-invasive NIRS probes have also been investigated in pregnant women. Japanese researchers measured placental oxygenation during pregnancy by placing probes on maternal abdominal surface [16–19]. They found higher placental tissue oxygenation in pregnancies complicated by hypertensive disorders and intrauterine fetal growth restriction (IUGR). Use of non-invasive NIRS monitoring of the placental oxygenation during labour has, however, not been studied yet.

The aim of our study was to determine whether changes in placental NIRS values during labour could identify intrapartum fetal hypoxia and resulting acidosis.

## Materials and methods

### Study participants

Women admitted to labour and delivery unit of the Department of Perinatology, University Medical Center Ljubljana, Slovenia in active first stage of labour were included in this prospective cohort study between September 2017 and March 2019.

Inclusion criteria were: singleton pregnancy, cephalic presentation, active phase of first stage of labour (regular contractions at least every 5 minutes with cervical change), gestational age between 37 and completed 41 weeks. Calculation of gestational age was based on the last menstrual period or determined by ultrasound (calculated by crown-rump length measured within the first trimester) when ultrasonographic estimation differed by $\geq$ 7 days from the one calculated by menstrual period. We only included women with anterior and fundal placental position.

Exclusion criteria were: suspected IUGR, oligohydramnios, polyhydramnios, diabetes mellitus (preexisting or gestational) requiring insulin treatment, and preeclampsia. Depth of subcutaneous tissue (from the skin surface to anterior uterine wall) measured by ultrasound > 5 cm was also an exclusion criterion.

Maternal and pregnancy information (age, parity, gestational age, body mass index at onset of pregnancy and at delivery, complications during pregnancy, smoking status, medications used) and information on course of labour (Bishop score at admission, medications applied, duration of labour, mode of delivery) were promptly entered into pre-established forms. At delivery, blood was sampled from the umbilical artery for acid-base analysis. Umbilical artery pH was chosen as pre-specified neonatal outcome.

All women provided written informed consent for study participation. The National Medical Ethics Committee approved the study on 14[th] March 2017 (reference number: 0120-65/2017-3; KME 60/03/17).

## CTG and NIRS measurements and analysis

Measurements were performed by one operator (KR) throughout the 18-month research period, both during day and night shifts to avoid patient selection bias. Measurements were carried out continuously until the end of the second stage of labour with the exception of possible visits to bathroom facilities, administration of epidural analgesia or need for operative delivery by emergency Caesarean section.

Electronic CTG recordings were made using the Avalon FM30 (Philips, Netherlands) apparatus with paper speed of 1 cm/min. Data were sampled at a frequency of 0.25 seconds and archived as computer records. The outputs of Avalon FM 30 include FHR and uterus activity. NIRS probe was attached to the maternal abdominal surface at the position just above the placenta (uterine fundus or anterior uterine wall), which was determined by ultrasound. The distance from the optode to the anterior border and to the centre of the placenta was documented. Nonin Equanox 7600 regional oximetry system (Nonin Medical, Inc., Plymouth, MN, USA) and electrode model Sensmart sensor 8004CA (Nonin Medical, Inc., Plymouth, MN, USA) with interoptode distance of 20 and 40 milimeters were used for all NIRS measurements (Figs 1 and 2). Data were sampled at a frequency of 1 second. The output of Nonin Equanox 7600 is regional tissue oxygen saturation ($SO_2$ –the percent of placental oxygenated blood). The light absorption information collected by the dual emitter/detector electrodes and transferred via sensor cables is automatically incorporated into Nonin's Dynamic Compensation light processing algorithm, which provides real time oxygenation saturation values of tissue examined. Recordings were later transferred to a computer for further analysis.

CTG analysis was performed by two qualified obstetricians (KR and ML) who were unaware of patient's NIRS results or birth outcomes at the time of analysis. The number of late decelerations (symmetrical gradual decrease and return of FHR, with nadir occurring after the peak of contraction)—considered reflex fetal responses to fetal hypoxia during contraction—

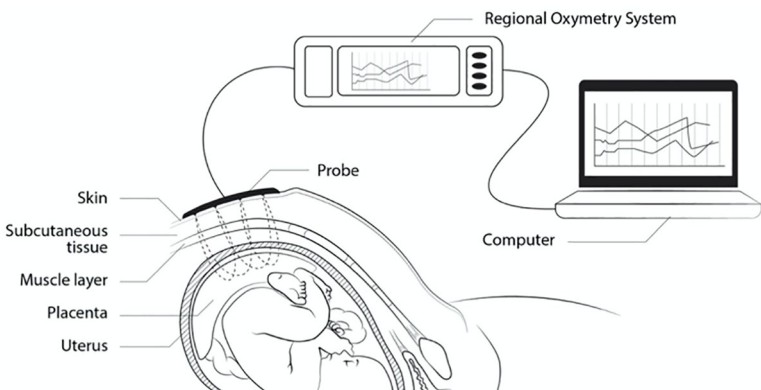

**Fig 1. Schematic representation of NIRS measurement.**

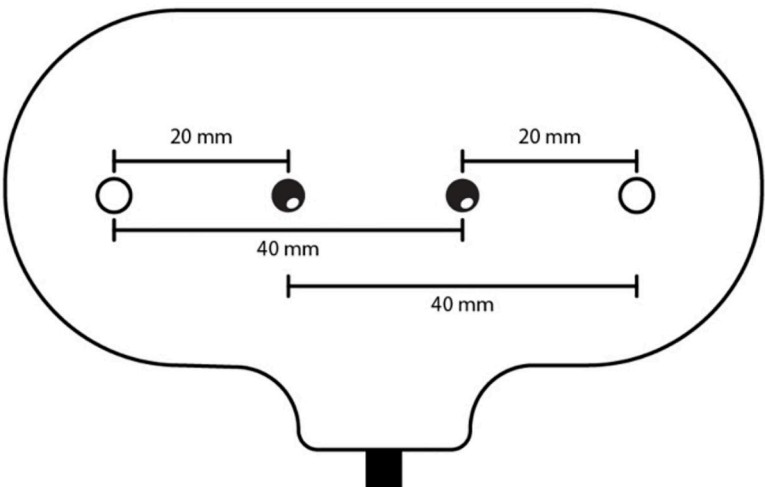

**Fig 2. NIRS electrode two light emitters and two detectors to provide measurements that are minimally affected by intervening tissues or surface effects.**

were analyzed in each CTG recording. We also analyzed the number of variable decelerations (abrupt decrease in FHR to levels below the baseline, which may occur in isolation or vary in onset, depth and duration in relationship to uterine contractions)–considered a sign of transient interruption of oxygen delivery to the fetus due to umbilical cord compression. We seperately analysed prolonged decelerations (decrease in FHR to levels below the baseline that lasts at least 2 minutes), which are believed to indicate a fetal chemo-receptor response to hypoxemia. Although early decelerations (symmetrical, gradual decrease and return of FHR below the baseline; in most cases the onset, nadir, and recovery of the deceleration is coincident with the beginning, peak, and ending of the contraction, respectively) are not considered related to fetal oxygenation, we chose to analyze also this type of decelerations and report their numbers. The overall number of decelerations within a record was also reported.

Fig 3 shows different subtypes of FHR decelerations as seen on CTG recording.

The variability of each CTG trace was assessed. Low variability was considered when duration of variability of < 5 bpm/min exceeded 50 minutes. In addition to evaluating these specific CTG parameters, the CTG tracing as a whole was assessed according to most commonly used CTG classification systems, i.e. those proposed by the International Federation of Gynecology and Obstetrics (FIGO), National Institute for Health and Care Excellence (NICE), American College of Obstetricians and Gynecologists (ACOG) and the classification proposed by Parer and Ikeda [20–23].

NIRS recordings were analyzed by studying episodes of placental deoxygenation. We used the decrease of ≥ 5% from baseline placental oxygenation to determine events of placental deoxygenation. We limited the duration of events to ≥ 15–180 seconds. A shorter duration could represent an artefact, whilst a duration exceeding 180 seconds was considered a baseline shift in tissue oxygenation. Fig 4 shows examples of NIRS occurring events.

Since the velocity of tissue deoxygenation has been shown to be an important parameter in critically ill patients, we also chose to analyze this NIRS parameter [24–27]. Tissue oxygenation velocity was calculated as the derivation of oxygenation, which is implemented as the change of oxygenation in one time sample of NIRS signal (% per second).

CTG and NIRS measurement systems have different recording formats as well as different sampling rates. Data processing was pursued in Matlab software, version R2016b (MathWorks,

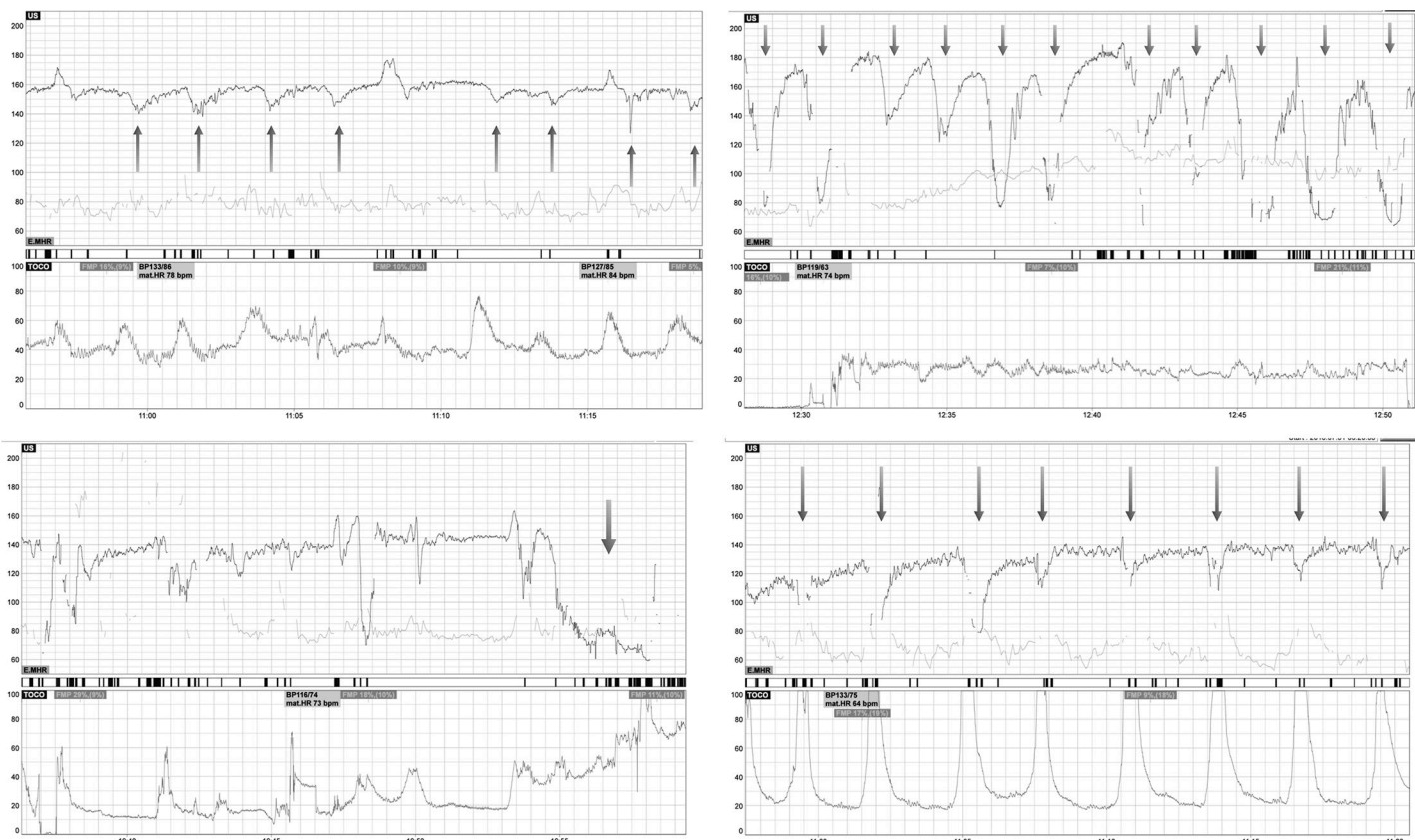

**Fig 3. Examples of different FHR deceleration subtypes on CTG.** a: Late FHR decelerations. b: Variable FHR decelerations. 3c: Prolonged FHR deceleration. c: Early FHR decelerations.

Natick, MA, USA) and converted to analysis-convenient data vectors without loss of information.

## Statistical analysis

CTG and NIRS parameters in group with vs. without neonatal pH $\leq$ 7.20 were compared using Mann-Whitney-U test. Categorical variables were compared using Chi-square test.

Receiver-operating-characteristics (ROC) curves were used to estimate predictive value of CTG and NIRS parameters that were shown to be significantly associated with neonatal umbilical artery pH $\leq$ 7.20. The program IBM SPSS Statistics for Macintosh, version 23.0 (IBM Corp, Armonk, NY, USA) was used for statistical comparison of groups and ROC analysis. Statistical significance was determined at $p < 0.05$.

In addition to classical ROC analysis, a computer-based statistical classification was also performed in Matlab (version R2016b; MathWorks, Natick, MA, USA) to further evaluate predictive values of CTG and NIRS for neonatal acidosis. Data were divided into two datasets: the first used for training and regressor selection, while the second for testing the obtained statistical classifier. This allowed assessing consistency of the results. Regressors were selected with backward elimination method, systematically eliminating one by one until the cross-validation results confirmed the minimal set of regressors that provides the classification accuracy of richer set of regressors [28]. Different classification methods were also tested using the Classification Learner application of Matlab [29].

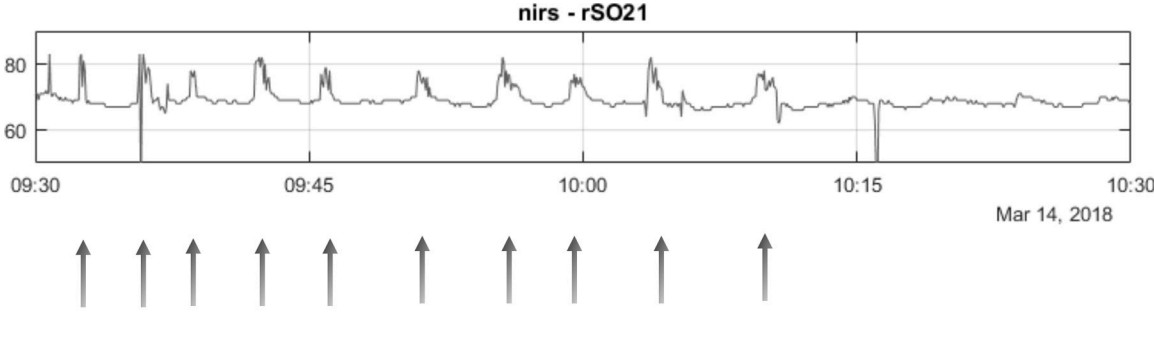

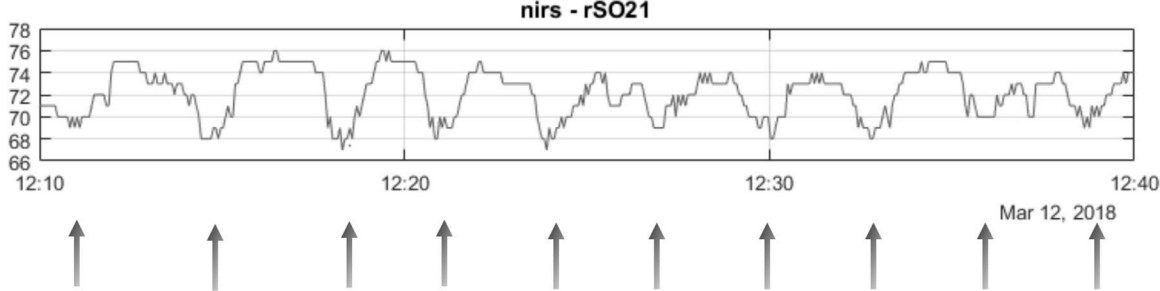

**Fig 4. Schematic representation of NIRS occurring events.** a: Rises in oxygen tissue saturation. b: Tissue desaturations.

## Results

CTG and NIRS were measured in 50 participants. Four measurements were excluded due to poor NIRS signal quality, umbilical cord blood sample was not available in three neonates, leaving 43 participants for analysis.

The age of participating women was between 18 and 48 years (median 31.0), gestational age between 38.4 and 41.4 weeks (median 40.1). Their median body mass index at onset of pregnancy was 22.1 (range 18.8 to 40.5) and 27.7 (range 23.2 to 41.7) at time of delivery. Eleven (26%) participants had anemia, treated with oral iron supplements. Nineteen women (44%) were nulliparous, the number of previous deliveries ranged from 1 to 6 in multiparous participants. Four women (9%) were regular smokers during pregnancy. There were no fetal or placental anomalies in the analyzed group. Median Bishop score at study inclusion was 6 (range 1–10). Placenta was located in the uterine fundus in 7 (16%) and anteriorly in 36 (84%) participants. The median distance from the probe to the anterior border and to the center of placenta was 15 mm (5–26 mm) and 35 mm (20–60 mm), respectively. Table 1 presents patient demographics in regard to umbilical artery pH value. Results show that pH $\leq$ 7.20 of the newborn was related to the mother being a nullipara, whilst smoking was a protective factor. No other patient demographic was shown to be statistically significantly related to the observed pH outcome.

Median duration of labour was 3.5 hours (0.25–9 hours). Seventeen women (39%) opted for epidural analgesia during labour. Six (14%) women gave birth via operative delivery (five emergency Cesarean sections and one vacuum extraction were performed due to fetal distress, diagnosed according to CTG– 5 (83%) of these neonates had umbilical artery pH $\leq$ 7.20). Five (12%) neonates were born with umbilical artery pH $\leq$ 7.20 with no intrapartum signs of suspicious or pathological CTG.

Table 2 shows CTG and NIRS parametres, that were significantly higher in the groups with vs. without neonatal pH $\leq$ 7.20. According to CTG classification by FIGO, 22 (51%) tracings

**Table 1. Patient clinical background variables in regard to umbilical artery pH value.**

| Variable | pH ≤ 7.20 (N = 10) | pH > 7.20 (N = 36) | p |
|---|---|---|---|
| Maternal age (years) | 29 (21–36) | 31 (18–48) | 0.42 |
| Nulliparous | 8 | 13 | 0.008 * |
| Parity | 1 (1–2) | 2 (1–6) | 0.001 * |
| Gestational age | 40 4/7 (39 0/7-41 4/7) | 40 0/7 (38 4/7-41 4/7) | 0.11 |
| BMI before pregnancy (kg/m$^2$) | 21.1 (18.8–32.6) | 23.1 (19.0–40.5) | 0.21 |
| BMI at delivery (kg/m$^2$) | 26.7 (24.1–36.8) | 28.0 (23.2–41.7) | 0.33 |
| Bishop at admission | 5 (3–9) | 6 (1–10) | 0.47 |
| Smoker | 0 | 4 | 0.04 * |
| Labour duration | 5.3 (0.5–7.0) | 3.0 (0.25–9.0) | 0.12 |
| d1 | 15 (5–25) | 15 (5–26) | 0.93 |
| d2 | 33 (21–55) | 36 (20–60) | 0.70 |

Data are shown as median (range) or N. P value was calculated by Mann-Whitney U-test and Chi square test.

* represents statistical significance (p<0.05).

BMI: body mass index, d1: distance from electrode to placenta, d2: distance from electrode to centre of placenta

were considered suspicious—of these, seven (31.8%) had pH ≤ 7.20, whilst fifteen (68.2%) had a normal pH. Seven (16%) traces were considered pathological; one (14.3%) had pH ≤ 7.20 and six (85.7%) normal pH. Fourteen tracings were considered normal. According to NICE CTG guidelines three (7%) tracings were suspicious; one (33.3%) had pH ≤ 7.20 and two (66.6%) normal pH. Eighteen (42%) tracings showed pathological characteristics; five (27.8%) neonates had pH ≤ 7.20 and thirteen (72.2%) normal pH. Twenty-two tracings were considered normal. Using ACOG guidelines, no tracings were considered pathological (category III), while the majority (95%) were suspicious (category II). Of these, ten (24.4%) neonates had pH ≤ 7.20 and thirty-one (75.6%) normal pH. Following the Parer&Ikeda classification system, eight tracings (19%) were given the blue, one (2%) yellow and one (2%) the orange color score. Two neonates (25%) color coded blue had pH ≤ 7.20 and 6 (75%) normal pH, one yellow coded neonate had a normal pH and one orange-coded had pH ≤ 7.20. The remaining 33 tracings were given the green colour score. Differences in incidence of suspicious and pathological CTG traces, regardless of classifications used, were not shown to be statistically significantly different among the groups with vs. without neonatal pH ≤ 7.20.

**Table 2. Cardiotocographic (CTG) and near infrared spectroscopy (NIRS) parameters in regard to umbilical arterial pH value.**

| CTG/NIRS parameter | pH ≤ 7.20 (N = 10) | pH > 7.20 (N = 33) | p |
|---|---|---|---|
| Overall no. of decelerations | 25 (3–91) | 10 (0–60) | 0.02* |
| No. of variable decelerations | 9 (0–76) | 7 (0–42) | 0.29 |
| No. of late decelerations | 0 (0–50) | 0 (0–0) | 0.07 |
| No. of early decelerations | 3 (0–47) | 1 (0–25) | 0.14 |
| No. of prolonged decelerations | 2 (0–4) | 1 (0–3) | 0.04* |
| No. of CTG with low variability | 1 (10%) | 2 (6%) | 0.46 |
| Overall no. of placental deoxygenations | 9 (2–37) | 2 (0–65) | 0.001* |
| Velocity of placental deoxygenation (%/s) | 2.31 (0–22) | 1 (0–49) | 0.03* |

Data are shown as median (range) or n (%). P value was calculated by Mann-Whitney test.

* represents statistical significance (p<0.05).

ROC curves were constructed for CTG and placental NIRS parameters which were statistically correlated with neonatal pH ≤ 7.20 (Fig 5). The figure shows a greater diagnostic reliability of NIRS in predicting fetal acidosis. Total number of placental deoxygenations has the highest area under the curve (AUC) of all parameters analysed.

ROC curves were also constructed for CTG categories, assessed by different classification systems. AUC with 95% confidence interval (95% CI) for classification system FIGO was 0.54 (0.35–0.73), NICE 0.57 (0.37–0.77), ACOG 0.53 (0.33–0.73), Parer & Ikeda 0.55 (0.34–0.77) respectively.

For the computer-based statistical classification, the first dataset (used for training and regressor selection) contained recordings and derived variables of 36 participants, including eight cases of neonatal acidosis. The second dataset (used for testing the obtained statistical classifier) contained recordings and derived variables of seven participants, including two cases of neonatal acidosis. The analysis carried out was 4-fold cross-validation on the first dataset. Number of folds, i.e. four, was determined according to the number of cases of neonatal acidosis in the first dataset. Each division of subgroups should have contained at least one case of neonatal acidosis. Three subgroups of the first dataset were used for training the classifier and for the selection of top ranking regressors. For the remaining subgroup, the previously created classifier was applied with the same regressors as the training subgroups (Fig 6). Among different classification methods, classification tree or decision tree method was chosen [30]. Following the backward elimination method, we were left with only one regressor: number of placental deoxygenations on NIRS.

The result of 4-fold cross-validation was the score, i.e., accuracy, of 75%. The best trained classifier had 25% false positive rate and 93% true positive rate. The final test of the obtained classifier was its test on the second dataset that was not used for selection and training. The result of the decision tree classifier using the number of deoxygenations on NIRS on the second dataset was the score of 100%.

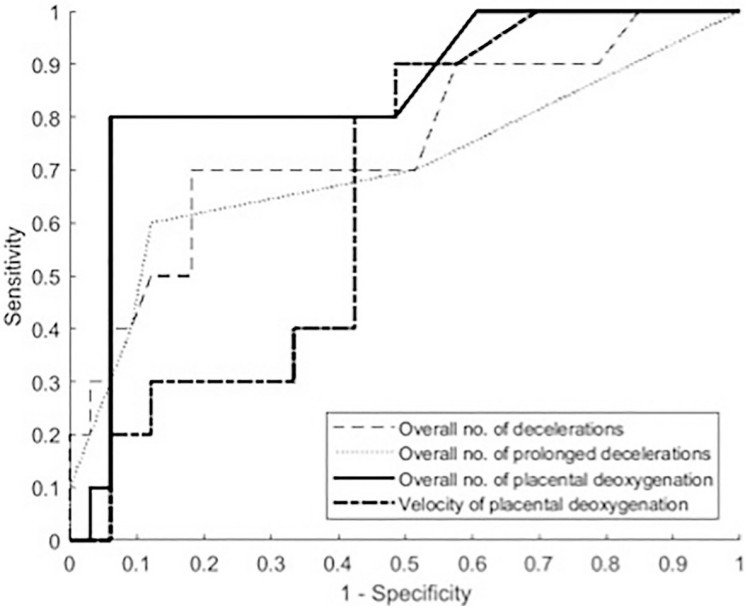

**Fig 5. Comparison of receiver-operating characteristics (ROC) curves for CTG and NIRS parameters to predict pH ≤ 7.20.** Overall no. of decelerations: AUC 0.75, 95% CI (0.57–0.94). Overall no. of prolonged decelerations: 0.71 (0.49–0.92). Overall no. of placental deoxygenation: 0.85 (0.70–0.99). Velocity of placental deoxygenation: 0.66 (0.49–0.83). AUC: area under the curve, CI: confidence interval.

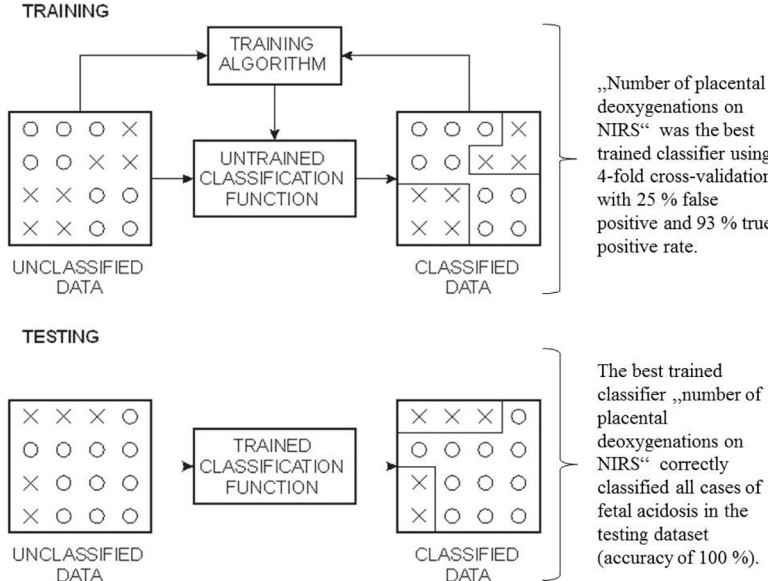

**Fig 6. Schematic illustration of computer-based statistical classification used for evaluation of predictive values of CTG and NIRS for neonatal acidosis.**

The training procedure used a training dataset with prior knowledge of correctly classified results for the computer-based algorithm which trains classification function (upper scheme). In our case, number of deoxygenations on NIRS provided the highest accuracy of predicting neonatal umbilical artery pH $\leq$ 7.20. The trained classification function was then used for the classification of test data (bottom scheme). Number of placental deoxygenations correctly classified all cases into pH $\leq$ 7.20 and pH > 7.20 groups (100% accuracy).

## Discussion

The aim of our study was to assess the applicability of non-invasive NIRS of the placenta for fetal surveillance during labour. We analysed specific CTG parametres and compared them to NIRS parameters. Since the interpretation of CTG is complex and is dependent also on other tracing characteristics (i.e. bradycardia, tachycardia, pathological prolonged decelerations), we included the assessment of the whole CTG tracing according to most commonly used CTG classification systems worldwide. Our results show that placental deoxygenations during labour measured by NIRS are associated with fetal/neonatal acidosis. Furthermore, predictive value of placental NIRS for neonatal acidosis was shown in both classic ROC analysis as well as in computer-based classification analysis to be superior to that of CTG, which is currently the gold standard for assessing fetal acid-base status during labour.

Despite the fact that NIRS is increasingly being adopted in different medical fields, studies on its potential uses in obstetrics and gynaecology (excluding anaesthesiologic care of the mother) are scarce. First studies date back almost 30 years, when Peebles et al. [12,31] reported that oxygen dynamics could be measured during labour by measuring cerebral NIRS of the fetus. Aldrich et al. reported that late decelerations in the CTG recording, as well as prolonged and variable decelerations, reflect inadequate fetal oxygenation and are associated with transient deoxygenation of the fetal brain [14,15]. The probes used in these studies were placed directly on the fetal head, making this a relatively invasive methodology. The drawbacks in

these studies also included probe movement with labour progression, changes in maternal position and pressure on the probe during uterine contractions, which could alter the interoptode space, leading to a change in the NIRS optical path length and inaccurate readings [32]. Poor contact between the optodes and fetal scalp (rapid descent of fetal head, body fluids impeding suction) was also problematic, as were numerous artefacts in the recordings. As a result, intrapartum fetal cerebral NIRS was not adopted into everyday obstetric practice.

Almost a decade later, four studies assessing placental oxygenation dynamics using transabdominal NIRS during pregnancy were published [16–19]. Kawamura et al. reported higher placental tissue oxygenation in pregnancies with IUGR, while found no association between placental oxygenation and gestational age [19]. Kakogawa et al. later described higher placental oxygenation also in pregnancies with gestational hypertension [16]. While all these studies were performed in pregnant women who were not in labour, our results could be viewed as being in contrast with these findings. Both IUGR and hypertensive disorders in pregnancy are well known factors of placental insufficiency which, according to these studies, could be associated with higher placental oxygenation. On the contrary, we found episodes of intrapartum deoxygenations of placental tissue to be associated with fetal acidosis. Hasegawa et al. suggested that placental oxygenation value depends on the aetiology of IUGR (decreased in umbilical abnormalities and increased in cases with placental abnormalities or preeclampsia) [18]. We did not include women with signs of chronic placental insufficiency in the present study, so further research will be needed to analyse baseline and intrapartum changes of placental NIRS in this population of parturitients.

Placental deoxygenation was defined as a decrease in NIRS value of $\geq 5\%$ for a period of $\geq 15$–180 seconds. Other fields of medicine generally use a preset cut-off for recognition of significant tissue desaturation. In our study, we could not use such a cut-off since this is a pilot study and there are no published data that could guide such a decision.

Small number of women included resulted in few severe cases of neonatal acidosis and no case of hypoxic ischaemic encephalophaty. Larger studies will be needed to determine if placental NIRS could also be a predictive measure for these severe adverse outcomes. Our study should, therefore, be viewed as "proof of concept" that measuring placental NIRS during labour is feasible and that these measurements could provide important information on fetal status. An important drawback of intrapartum placental NIRS is the possibility that women with posteriorly located placentas would not benefit from such additional monitoring methodology since scattering and absorption could affect the detection of changes in the near-infrared light. This is why we chose not to include women with posterior placental location, but this hypothesis will have to be confirmed in further research. Even if confirmed, however, placental NIRS measurements would still be feasible in a great proportion of the population since the prevalence of posterior placentas is around 25% [33]. Another limitation of this non-invasive monitoring method could be its potentially lower accuracy in obese women. Our results do not confirm this. Of all women screened for depth of subcutaneous tissue, we found no woman with more than 5 cm of subcutaneous tissue, which was chosen as an exclusion criterion due to limitations in the depth of NIRS analysis.

## Supporting information

**S1 Data.**
(XLSX)

**S2 Data.**
(XLSX)

**S3 Data.**
(XLSX)

**S4 Data.**
(XLSX)

## Author Contributions

**Conceptualization:** Matej Podbregar, Miha Lučovnik.

**Data curation:** Juš Kocijan.

**Formal analysis:** Juš Kocijan.

**Investigation:** Katja Ražem.

**Methodology:** Matej Podbregar, Miha Lučovnik.

**Project administration:** Katja Ražem.

**Resources:** Miha Lučovnik.

**Software:** Juš Kocijan.

**Supervision:** Miha Lučovnik.

**Validation:** Juš Kocijan, Matej Podbregar.

**Visualization:** Katja Ražem, Juš Kocijan.

**Writing – original draft:** Katja Ražem.

**Writing – review & editing:** Katja Ražem, Juš Kocijan, Matej Podbregar, Miha Lučovnik.

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
