## [Decision Letter · Decision Letter 0]

9 Jan 2020

PONE-D-19-25139

Near-infrared spectroscopy of the placenta for monitoring fetal oxygenation during labour

PLOS ONE

Dear Ražem,

Thank you for submitting your manuscript to PLOS ONE. After careful consideration, we feel that it has merit but does not fully meet PLOS ONE’s publication criteria as it currently stands. Therefore, we invite you to submit a revised version of the manuscript that addresses the points raised during the review process.

We would appreciate receiving your revised manuscript by Feb 23 2020 11:59PM. To enhance the reproducibility of your results, we recommend that if applicable you deposit your laboratory protocols in protocols.io, where a protocol can be assigned its own identifier (DOI) such that it can be cited independently in the future. For instructions see: http://journals.plos.org/plosone/s/submission-guidelines#loc-laboratory-protocols

We look forward to receiving your revised manuscript.

Kind regards,

Anna Palatnik, M.D.

Academic Editor

PLOS ONE

Journal Requirements:

2. In your Methods section, please provide additional information about the participant recruitment method and the demographic details of your participants.

Please ensure you have provided sufficient details to replicate the analyses such as: a) a table of relevant demographic details and b) a description of how participants were recruited.

Reviewers' comments:

Reviewer's Responses to Questions

**Comments to the Author**

1. Is the manuscript technically sound, and do the data support the conclusions?

Reviewer #1: Partly

2. Has the statistical analysis been performed appropriately and rigorously? 

Reviewer #1: I Don't Know

3. Have the authors made all data underlying the findings in their manuscript fully available?

Reviewer #1: Yes

4. Is the manuscript presented in an intelligible fashion and written in standard English?

Reviewer #1: Yes

5. Review Comments to the Author

Reviewer #1: This work describes an application of NIRS for monitoring the oxygenation state of fetuses during labour. Cardiotocography (CTG) is the standard method for predicting neonatal acidosis but it is characterized by a low positive predictive value and by large inter- intra operator variability for the interpretation of the CTG traces. In this study, 43 pregnant women were divided in two groups, 10 resulting on children having PH <7.2 and 33 having a PH > 7.2 at birth. Several parameters from CTG and NIRS were used to discriminate these two groups and for their predictive values for children acidosis, by standard ROC curves or by a machine learning based method. The method having the best predictive accuracy is based on the number of deoxygenation episodes measured by NIRS.

The paper shows indeed an original application of NIRS, which is used non-invasively (unlike previous works in the literature with the same application) for monitoring the fetuses’ oxygenation during labour. While the abstract is concise and well written, I think that some sections of the paper should be improved for the paper to be easily accessible to a broader audience. Especially I think that the paper would benefit of more figures (and/or schematics).

For example, the section on CTG and NIRS measurements and analysis should be expanded to provide more details. What are the outputs of Avalon FM 30 (other than FHR and uterus contraction) and Nonin Equanox 7600? More information (possibly literature references or a schematic) should be provided for the NIRS machine, like number of sources and detectors and principle of data analysis (saying that the instrument comprises two inter-optode distances does not describe fully the instrument). Does it measure the changes of oxy-, deoxy and total hemoglobin only, or also tissue oxygen saturation? How are the data at different source-detector distances combined? Also, in the same section the authors describe different CTG “deceleration” (probably heart rate deceleration), like “late”, “variable”, “prolonged”, “early” which are not easily grasped by non-medical audience. For NIRS measurements, the authors arbitrarily defined episodes of deoxygenation as those where there was a >5% decrease on placenta oxygenation (from baseline values) and lasting a time range of (15, 80) s. All this information could enormously benefit if it was complemented by plots of typical experimental results both for CTG and NIRS. The authors should include one or more examples of CTG plots where the different decelerations are marked; also, for NIRS the authors should add typical traces of the parameters being measured and the episodes of deoxygenation marked. It is unclear which parameter was measured for the change in oxygenation: a) decrease of oxy-hemoglobin concentration; b) increase in deoxy-hemoglobin concentration; c) both; d) decrease of tissue oxygen saturation. How does a typical NIRS recording looks like? I am concerned, given the type of application for the presence of motion artifacts that could affect the interpretation of the data. The authors describe changes of oxygenation that last more than 3 minutes as baseline shifts of tissue oxygenation, which are due I believe to motion artifacts. Also, motion artifacts could be present in the range (15,80) seconds. The authors should comment on these points and show typical traces.

How is the velocity of tissue deoxygenation measured from the traces?

About the section of statistical analysis, I wonder if the authors have tried to use different training datasets and therefore different testing datasets. In other words, why the test is done only on one data set?

Which classification methods were tested? Please add a reference for the classification learner approach.

About the results section, when the authors described the CTG classification according to FIGO, NICE, ACOG, etc., it seems that not all the 43 cases could be classified (for example 29 were classified by FIGO and 21 by NICE). This also should be explained.

In the discussion section the authors should try to address the critical issue of different choices for the definition of episodes of deoxygenation, with different thresholds for the changes and time duration. How would these choices affect the classification results? Is the choice of 5%, and especially the time range (15, 80) seconds a meaningful choice for the definition of the episodes of deoxygenation? Is this choice guided by the temporal evolution of other parameters, like the duration of uterus contractions?

Minor issues and typos:

In the abstract when describing the results of prolonged decelerations, the symbol of the p-value “p” is missing (p=0.04).

Line 44: the symbols 37+0 and 41+6 should be explained.

Line 77: change “electrode” into “optode”.

In the result section actual numbers and the words referring to the numbers are inconsistently used. For example, line 152: “17” instead of “Seventeen”. See also line 202: “eight” instead of “8” and line 204 “two” instead of “2”.

Line 176: “spectroscopy” is misspelled.

Have the authors done a comprehensive literature search about the use of NIRS during labour? For example, I found the following review that is not cited: D. M. Peebles et al., “Fetal cerebral oxygenation and hemodynamics during labour measured by NIRS,” Mental retardation and developmental disabilities, 59-68 (1997).

6. PLOS authors have the option to publish the peer review history of their article (what does this mean?). If published, this will include your full peer review and any attached files.

Reviewer #1: No

---

## [Author Response · Author response to Decision Letter 0]

31 Jan 2020

RESPONSES TO TO THE EDITOR AND REVIEWER:

RESPONSES TO THE EDITOR:

We have followed the journal’s requirements.

2. In your Methods section, please provide additional information about the participant recruitment method and the demographic details of your participants.

Please ensure you have provided sufficient details to replicate the analyses such as: a) a table of relevant demographic details and b) a description of how participants were recruited.

Table 1 presenting patients demographics has been added and the table with CTG and NIRS parameters renamed accordingly. We believe inclusion and exclusion criteria are presented in sufficient detail to allow replication of our work.

We are willing to share our data since there are no legal or ethical restriction for doing so. Tables with de-identified data have now been uploaded together with the revised manuscript, our responses and the revised cover letter.

Data have now been uploaded.

RESPONSES TO THE REVIEWER:

INTRODUCTION:

The section on CTG and NIRS measurements and analysis should be expanded to provide more details:

- What are the outputs of Avalon FM 30 (other than FHR and uterus contraction) and Nonin Equanox 7600? 

Thank you for your comment. We have incorporated the outputs analysed into the manuscript for better clarification:

The outputs of Avalon FM 30 are FHR and uterus activity, the output of Nonin Equanox 7600 is regional tissue oxygen saturation (SO2 – the percent of placental oxygenated blood). 

- More information (possibly literature references or a schematic) should be provided for the NIRS machine, like number of sources and detectors and principle of data analysis (saying that the instrument comprises two inter-optode distances does not describe fully the instrument). 

All authors agree with this comment and have therefore provided the following schematic representations:

Figure 1: Schematic representation of NIRS measurement

Figure 2: NIRS electrode used in our study. Two light emitters and two detectors provide measurements that are minimally affected by intervening tissues or surface effects

The light absorption information collected by the dual emitter/detector electrodes and transferred via sensor cables is automatically incorporated into Nonin’s Dynamic Compensation light processing algorithm, which provides real time oxygenation saturation values of tissue examined.

Recordings were later transferred to a computer for further analysis.

- Does it measure the changes of oxy-, deoxy and total hemoglobin only, or also tissue oxygen saturation? How are the data at different source-detector distances combined? 

Thank you for your comment.

It measures tissue oxygen saturation, the detailed algorithm of Nonin is, however, proprietary.

Several NIRS devices are available for clinical use, which differ according to numerous aspects, including algorithms adopted, type of light source, wavelengths of light emitted, the number and distance between the light emitters and detectors (Bickler PE, Feiner JR, Rollins MD. Factors affecting the performance of 5 cerebral oximeters during hypoxia in healthy volunteers. Anesth Analg. 2013;117(4):813-23., 

Kovač P, Miš K, Pirkmajer S, Marš T, Klokočovnik T, Kotnik G, Podbregar E, Podbregar M. How to Measure Tissue Oxygenation Using Near-Infrared Spectroscopy in a Patient With Alkaptonuria. J Cardiothorac Vasc Anesth. 2018 Dec;32(6):2708-2711.).

The Nonin Equanox device uses a dual light emitting and detecting sensor architecture, which means that the measurements are less affected by intervening tissue or surface effects. Use of four wavelengths of NIR light (730 nm, 760 nm, 810 nm, 880 nm) increases the accuracy of reporting the actual percent of oxygenated hemoglobin in the targeted tissue and also allows the algorithm to reduce inter-subject variability, regardless of age weight or skin color.

- CTG decelerations “late”, “variable”, “prolonged”, “early” are not easily grasped by non-medical audience. 

Thank you for pointing this out. The authors agree with your comment and have therefore added the definitions of terms into the manuscript, illustrated with examples.

Late decelerations (symmetrical gradual decrease and return of FHR, with nadir occurring after the peak of contraction) are considered reflex fetal responses to fetal hypoxia during contractions.

CTG 1: Late FHR decelerations

Variable decelerations (abrupt decrease in FHR to levels below the baseline, which may occur in isolation or vary in onset, depth and duration in relationship to uterine contractions) are considered a sign of transient interruption of oxygen delivery to the fetus due to umbilical cord compression. 

CTG 2: Variable FHR decelerations

Prolonged decelerations (decrease in FHR to levels below the baseline that lasts at least 2 minutes), are believed to indicate a fetal chemo-receptor response to hypoxemia. 

CTG 3: Prolonged FHR deceleration 

Early decelerations (symmetrical, gradual decrease and return of FHR below the baseline. In most cases the onset, nadir, and recovery of the deceleration are coincident with the beginning, peak, and ending of the contraction, respectively) are not considered related to fetal oxygenation.

CTG 4: Early FHR decelerations

- For NIRS measurements, the authors arbitrarily defined episodes of deoxygenation as those where there was a >5% decrease on placenta oxygenation (from baseline values) and lasting a time range of (15, 180) s. All this information could enormously benefit if it was complemented by plots of typical experimental results both for CTG and NIRS: include one or more examples of CTG plots where the different decelerations are marked; also, for NIRS the authors should add typical traces of the parameters being measured and the episodes of deoxygenation marked. 

Similarly to the comment above, we have included schematic examples of NIRS occurring events into the manuscript. We believe this does indeed equip the reader with a better understanding of described events.

NIRS 1: Rises in oxygen tissue saturation 

NIRS 2: Desaturations 

The examples of decelerations on CTG are shown above (CTG 1-4)

- It is unclear which parameter was measured for the change in oxygenation: a) decrease of oxy-hemoglobin concentration; b) increase in deoxy-hemoglobin concentration; c) both; d) decrease of tissue oxygen saturation. 

Tissue oxygenation value was measured for observing change in oxygenation. 

In the methods section, we describe how the light absorption information is collected by the dual emitter/detector electrodes and transferred via sensor cables, which provides real time oxygenation saturation values of tissue examined.

-How does a typical NIRS recording look like? I am concerned, given the type of application for the presence of motion artifacts that could affect the interpretation of the data. The authors describe changes of oxygenation that last more than 3 minutes as baseline shifts of tissue oxygenation, which are due I believe to motion artifacts. Also, motion artifacts could be present in the range (15,180) seconds. The authors should comment on these points and show typical traces. 

Short term motion artefacts are present throughout the whole monitoring period, which is why a pre-specified 15 second threshold was chosen to limit their impact. No rise in in tissue oxygenation is documented in example 1, since the change is smaller than 5 % from the baseline. In example 2, no fall in desaturation is documented, since the events last less than 15 seconds each. In both cases, the threshold is not reached for defining occurring events - therefore limiting signal artefacts.

EXAMPLE 1 

EXAMPLE 2 

In our study, we believe that a > 3 min oxygenation change represents a new, stable, basal oxygenation level. In our opinion absolute changes in measurements are more indicative than arbitrary preset cut-off values. Example 3 shows three different average basal saturation values, each lasting more that 3 min each.

EXAMPLE 3

- How is the velocity of tissue deoxygenation measured from the traces? 

Thank you for the question. For better understanding, we have added this explanation to the manuscript:

Tissue oxygenation velocity is calculated as the derivation of oxygenation, which is implemented as the change of oxygenation in one time sample of NIRS signal (% per second). 

STATISTICAL ANALYSIS: 

-Have authors tried to use different training datasets and therefore different testing datasets. In other words, why the test is done only on one data set?

Data were divided into two datasets: the first used for training and regressor selection, while the second for testing the obtained statistical classifier. For the computer-based statistical classification, the first dataset (used for training and regressor selection) contained recordings and derived variables of 36 participants, including eight cases of neonatal acidosis. The second dataset (used for testing the obtained statistical classifier) contained recordings and derived variables of seven participants, including two cases of neonatal acidosis. The analysis carried out was 4-fold cross-validation on the first dataset. Number of folds, i.e. four, was determined according to the number of cases of neonatal acidosis in the first dataset. Each division of subgroups should have contained at least one case of neonatal acidosis. Three subgroups of the first dataset were used for training the classifier and for the selection of top ranking regressors. For the remaining subgroup, the previously created classifier was applied with the same regressors as the training subgroups. More than 4-fold divison of the first set and more than one testing dataset were not possible due to the limited number of available measurements. Among different classification methods, classification tree or decision tree method was chosen (Grochtmann M, Grimm K. Classification trees for partition testing. Software Testing, Verification and Reliability. 1993;3: 63–82. doi:10.1002/stvr.4370030203)

- Which classification methods were tested? Please add a reference for the classification learner approach. 

Different classification methods were also tested using the Classification Learner application of Matlab (reference: Statistics and Machine Learning Toolbox User’s Guide, Mathworks, Natick, MA, 2016).

This has been added to the manuscript.

RESULTS: 

- About the results section, when the authors described the CTG classification according to FIGO, NICE, ACOG, etc., it seems that not all the 43 cases could be classified (for example 29 were classified by FIGO and 21 by NICE). This also should be explained. 

Only the number of suspicious and pathological traces was mentioned in the text, the others were considered normal. To eliminate confusion, we have included this into the manuscript. Thank you for pointing this out.

DISCUSSION: 

-Try to address the critical issue of different choices for the definition of episodes of deoxygenation, with different thresholds for the changes and time duration. How would these choices affect the classification results? Is the choice of 5%, and especially the time range (15, 180) seconds a meaningful choice for the definition of the episodes of deoxygenation? Is this choice guided by the temporal evolution of other parameters, like the duration of uterus contractions? 

Our study was the first study analyzing placental NIRS values during labour. Therefore, we had no previously published data on which we could base our outcome definitions on. 

We defined an occurring event on NIRS as a change of ≥ 5 % for a period of ≥ 15 -180 seconds. Understandably, a shorter duration and smaller change would result in more occurring events, however, we believe these would more frequently include artefacts (movement of patient, potential loss of signal...). Additionally, occurring CTG events have similar durations; definitions of accelerations and decelerations include at least 15 second changes of fetal heart rate, whilst the duration of prolonged decelerations according to FIGO and NICE classification systems exceeds 180 seconds. A change of ≥ 5 % was chosen based on other NIRS studies in non-pregnant populations and since we assumed such change could be of clinical importance.

As mentioned, this was a pre-specified outcome. We agree with the reviewer, that different NIRS parameters and different cut-offs could yield different (potentially even better) prognostic values as ones reported in our study. The study, therefore, highlights the need for further research in this field.

How would different episode definitions affect classification results? 

As mentioned above, a change in signal > 5 % was a pre-specified outcome, determined before performing analysis of data. Further studies which would apply different specified episodes could show different (perhaps even better) results. Since our study was the first of this kind, a 5 % was considered an educated guess.

Minor issues and typos: 

Thank you for your suggestions. These have all been corrected and inserted into the manuscript, along with the suggested reference.

- In the abstract when describing the results of prolonged decelerations, the symbol of the p-value “p” is missing (p=0.04).

- Line 44: the symbols 37+0 and 41+6 should be explained.

- Line 77: change “electrode” into “optode”.

- In the result section actual numbers and the words referring to the numbers are inconsistently used. For example, line 152: “17” instead of “Seventeen”. See also line 202: “eight” instead of “8” and line 204 “two” instead of “2”.

- Line 176: “spectroscopy” is misspelled.

- Have the authors done a comprehensive literature search about the use of NIRS during labour? For example, I found the following review that is not cited: D. M. Peebles et al., “Fetal cerebral oxygenation and hemodynamics during labour measured by NIRS,” Mental retardation and developmental disabilities, 59-68 (1997)

---

## [Decision Letter · Decision Letter 1]

25 Mar 2020

Near-infrared spectroscopy of the placenta for monitoring fetal oxygenation during labour

PONE-D-19-25139R1

Dear Dr. Ražem,

We are pleased to inform you that your manuscript has been judged scientifically suitable for publication and will be formally accepted for publication once it complies with all outstanding technical requirements.

With kind regards,

Anna Palatnik, M.D.

Academic Editor

PLOS ONE

Additional Editor Comments (optional):

All comments have been addressed

Reviewers' comments:

Reviewer's Responses to Questions

**Comments to the Author**

1. If the authors have adequately addressed your comments raised in a previous round of review and you feel that this manuscript is now acceptable for publication, you may indicate that here to bypass the “Comments to the Author” section, enter your conflict of interest statement in the “Confidential to Editor” section, and submit your "Accept" recommendation.

Reviewer #1: All comments have been addressed

2. Is the manuscript technically sound, and do the data support the conclusions?

Reviewer #1: Yes

3. Has the statistical analysis been performed appropriately and rigorously? 

Reviewer #1: Yes

4. Have the authors made all data underlying the findings in their manuscript fully available?

Reviewer #1: Yes

5. Is the manuscript presented in an intelligible fashion and written in standard English?

Reviewer #1: Yes

6. Review Comments to the Author

Reviewer #1: All the comments have been addressed. The figures have added more clarity to the manuscript and the paper is definitely ready for publication.

7. PLOS authors have the option to publish the peer review history of their article (what does this mean?). If published, this will include your full peer review and any attached files.

Reviewer #1: Yes: Angelo Sassaroli

---

## [Editor Report · Acceptance letter]

30 Mar 2020

PONE-D-19-25139R1 

Near-infrared spectroscopy of the placenta for monitoring fetal oxygenation during labour 

Dear Dr. Ražem:

I am pleased to inform you that your manuscript has been deemed suitable for publication in PLOS ONE. Congratulations! Your manuscript is now with our production department. 

With kind regards,

on behalf of

Dr. Anna Palatnik 

Academic Editor

PLOS ONE